# Effect of 6-Week Balance Exercise by Real-Time Postural Feedback System on Walking Ability for Patients with Chronic Stroke: A Pilot Single-Blind Randomized Controlled Trial

**DOI:** 10.3390/brainsci11111493

**Published:** 2021-11-12

**Authors:** Makoto Komiya, Noriaki Maeda, Taku Narahara, Yuta Suzuki, Kazuki Fukui, Shogo Tsutsumi, Mistuhiro Yoshimi, Naoki Ishibashi, Taizan Shirakawa, Yukio Urabe

**Affiliations:** 1Graduate School of Biomedical and Health Sciences, Hiroshima University, Hiroshima 734-8553, Japan; norimmi@hiroshima-u.ac.jp (N.M.); kazuki-fukui@hiroshima-u.ac.jp (K.F.); shogo-tutumi@hiroshima-u.ac.jp (S.T.); mitsuhiroyoshimi0116@hiroshima-u.ac.jp (M.Y.); yurabe@hiroshima-u.ac.jp (Y.U.); 2Department of Rehabilitation, Matterhorn Rehabilitation Hospital, Hiroshima 737-0046, Japan; t9toct82@yahoo.co.jp (T.N.); yt.suzuki28@gmail.com (Y.S.); rugby1216nn@gmail.com (N.I.); 3Department of Orthopedics, Matterhorn Rehabilitation Hospital, Hiroshima 737-0046, Japan; matter@jasmine.ocn.ne.jp

**Keywords:** chronic stroke, balance exercise, randomized controlled trial

## Abstract

Stroke causes balance dysfunction, leading to decreased physical activity and increased falls. Thus, effective balance exercises are needed to improve balance dysfunction. This single-blind, single-center randomized controlled trial evaluated the long-term and continuous effects of balance exercise using a real-time postural feedback system to improve balancing ability safely. Thirty participants were randomized into intervention (*n* = 15) and control (*n* = 15) groups; 11 in each group completed the final evaluation. The effect of the intervention was evaluated by muscle strength of knee extension, physical performance (short physical performance battery, the center of pressure trajectory length per second, and Timed Up and Go test [TUG]), and self-reported questionnaires (modified Gait Efficacy Scale [mGES] and the Fall Efficacy Scale) at pre (0 week), post (6-week), and at follow-up (10-week) visits. The TUG and mGES showed a significant interactive (group * time) effect (*p* = 0.007 and *p* = 0.038, respectively). The intervention group showed significant decreasing time to perform TUG from pre- to post-intervention (*p* = 0.015) and pre-intervention to follow-up (*p* = 0.016); mGES showed a significant change from pre-intervention to follow-up (*p* = 0.036). Thus, balance exercise using a real-time postural feedback system can confer a positive effect on the walking ability in patients with chronic stroke and increase their self-confidence in gait performance.

## 1. Introduction

Stroke is a major cause of death and disability worldwide [1,2] and influences physical, mental, and cognitive functions [3]. The most common physical dysfunction related to chronic stroke is impaired balance. Previous studies on the prevalence of balance impairment have reported an incidence of 61–83% [4,5]; even in the chronic phase, the incidence is as high as 22–43% [4,6]. Reduced ability to balance in patients with chronic stroke increases the risk of falls and social isolation and decreases physical activity [7]. Balance exercise for patients with chronic stroke has positive effects on postural control during walking and mobility, and this is a common approach to improving these disorders.

Balance exercises for patients with chronic stroke include the use of unstable surfaces [8], virtual reality [9], and aquatic therapy [10] as the somatosensory, visual, and vestibular system approaches, respectively. These are considered important in balance exercises after stroke [11,12]. When used in conjunction with general rehabilitation, these methods have been shown to improve posture control and walking ability even in patients with chronic stroke [8,9]. For example, the effect of intervention using visual feedback has been reported to show more improvement in postural sway and dynamic balance compared to conventional rehabilitation [13], and the auditory feedback has been suggested to help improve gait symmetry and foot pressure [14]. Similarly, somatosensory feedback is effective in improving posture control [15]. Thus, a variety of feedback-based exercises are conducted to improve the balancing ability of patients with stroke. These interventions can be performed safely and can be highly effective.

One of the essential aims of balance practice in patients with stroke is to alter the muscle response to postural changes and external stimuli and reweigh that sensation. This is a critical theory that directly relates to improving muscle activity abnormalities that occur during walking in patients with stroke to improve walking. The effect of balance learning on gait improvement by electrical stimulation for patients with stroke as an approach to muscle response was recently reported [16]. This suggests that adjusting the response of muscle activity to posture changes can positively affect muscle exertion during walking, leading to improved walking. In this study, we focused on a real-time postural feedback system as a way to achieve effective feedback and ensure safe balance exercises. The system is designed to reduce or increase the movement of the platform by up to 15% relative to the length of the trajectory of the center of pressure in real time, which can provide sensory feedback to an extent at a range of non-perceptual levels [17]. In addition, a previous study showed that the effect of the anti-phase mode of the same device used in the present study for healthy adults shortened the onset time of muscle activity in response to changes in center of pressure (COP) oscillations [17]. A study using this system showed that balance exercises improved walking ability in patients with Parkinson’s disease and spinocerebellar degeneration, and the effect was maintained for one week [18]. The results of this study suggested that balancing exercises with non-sensory movements of the platform could improve the walking ability of subjects with impaired motor coordination by changing their muscle response to postural changes. Hence, if research can demonstrate the effects of such balance exercises after stroke, it might be possible to provide a new type of balance exercise that safely improve postural control during walking and mobility. However, there is no evidence to show the effectiveness of balance exercises using a real-time postural feedback system for patients with chronic stroke.

Therefore, we aimed to compare the effects of balance exercise using a real-time position feedback system with those of conventional balance exercise performed under unstable conditions for long-term intervention (six weeks) and post-intervention sustained effects (after four weeks) in patients with stroke for clinical application. The hypotheses were: (1) the intervention effect using a real-time position feedback system would be higher than that of the conventional balance exercise on unstable surfaces; and (2) the effect obtained during the intervention period would be sustained after four weeks.

## 2. Materials and Methods

### 2.1. Study Design and Setting

A single-blind (patients), single-center randomized controlled trial (RCT) was conducted. Participants were recruited from the day-care center of the Matterhorn Rehabilitation Hospital in Kure City, Hiroshima Prefecture, Japan, and recruitment was completed for the intervention between June 2020 and January 2021. The day-care center treats patients who have passed the convalescent phase of rehabilitation but still require continuous rehabilitation due to functional disabilities such as gait or balance disorders. The sample size was calculated using G*power 3.1.9.2 software (Heinrich-Heine-University Düsseldorf, version 3.1.9.4, Düsseldorf, Germany). Before the present study, a pilot study to estimate the effect size was conducted to detect the sample size of this study as there was no previous study that used the Balance Adjustment System (BASYS) as an intervention and Timed Up and Go (TUG) to determine the intervention effect. The effect size was calculated by comparing the change in TUG after BASYS intervention in five patients with chronic stroke using paired *t*-test (pre vs. post) and was set at 0.25. Therefore, for this study, the effect size, mean power, and alpha error were set at 0.25, 0.8, and 0.05, respectively. The analysis using G*power software showed that at least 14 participants would make an acceptable sample size for each group; thus, 43 participants were recruited in consideration of dropouts.

The following inclusion criteria were used: (1) at least 12 months after an ischemic or hemorrhagic stroke, (2) Brunnstrom recovery stage III or higher, (3) ability to walk at least 10 m with or without assistive devices, and (4) ability to communicate verbally with sufficient understanding of the research purpose and methods and not have severe cognitive impairment. The exclusion criteria were as follows: (1) a history of treatment for orthopedic or musculoskeletal injuries affecting the lower extremities, and (2) a history of neurological dysfunction, such as seizure disorder, head injury, or peripheral neuropathy. A total of 43 participants were screened for eligibility; 13 participants did not meet the inclusion criteria. Thus, the 30 eligible participants were randomly assigned to two groups: intervention group (*n* = 15) and control group (*n* = 15). This was done by the research collaborator’s physical therapist using the “www.randomizer.org” (assessed on 4 May 2020) random sampling method. The participants were unaware of the group they had been assigned to. After random assignment, baseline measurements were taken. Then, after six weeks of intervention in each group, post-intervention measurements were taken. Follow-up measurements were taken four weeks later. The research protocol was approved by the Ethics Committee on clinical trials of the Matterhorn Rehabilitation Hospital (MHR19004). This RCT was registered at UMIN (UMIN000044), and we followed the guidelines issued by the Consolidated Standards of Reporting Trials (CONSORT) [19]. Before participating in the study, the contents of the study were explained to all participants, and they provided written, informed consent.

### 2.2. Intervention

The intervention was conducted during a rehabilitation session in the day-care center. The intervention group performed six weeks of balance exercises (twice a week, two 1-min sessions, for a total of 2 min per day) using the stabilometer with in-built disturbance generation (BASYS, Tec Gihan Co. Ltd., Kyoto, Japan); the control group used a polyurethane mat (46.0 cm (L) × 46.0 cm (W) × 6.0 cm (H), StimUp Balance Pad, Celcom, Inc., Fukuoka, Japan). Participants of the control group were instructed to only stand on the polyurethane mat for the same interval as the intervention group. All participants were instructed to maintain a static standing posture with their eyes open on each support surface. The stabilometer was used in the anti-phase mode, where the platform was moved through the inverse phase of the change in the COP displacement. For the first minute of each session, the platform swayed in the forward-backward direction; for the remaining 1 min, it swayed in the left-right direction (Figure 1). The amount of movement of the platform was set at 5% for the first two weeks; thereafter, it was increased by 5% every two weeks. In addition to the balance exercise, both groups received standard physical therapy twice a week, such as muscle strengthening exercises, stretching, and walking.

### 2.3. Outcome Measures

Each group was evaluated at three time points: at 0 weeks, 6 weeks, and 10 weeks, with muscle strength of knee extension, physical performance, and self-rated questionnaires for fall and gait efficacy. The TUG was set as a primary outcome along with other secondary outcome measures (muscle strength of knee extension, Short Physical Performance Battery (SPPB), the ability to maintain postural stability, and self-reported questionnaires for fall and gait efficacy).

#### 2.3.1. Demographic and Baseline Data

Demographic characteristics collected were age, sex, height, weight, and body mass index (BMI). The following baseline clinical characteristics were recorded: (1) use of a walking aid; (2) use of an ankle joint orthosis; (3) the time from the onset of stroke to the start of the intervention; (4) Revised Hasegawa Dementia Scale (HDS-R) score to screen the severity for cognitive impairment with a maximum score of 30; and (5) the Brunnstrom recovery stage for the lower limb.

#### 2.3.2. Muscle Strength of Knee Extension

The maximum isometric knee extensor strength was assessed using a handheld dynamometer (Mobie, Sakai Med Co., Tokyo, Japan) when participants were seated on a treatment table with their knees and hips at 90° flexion. The sensor pad was fixed to the distal lower leg with a Velcro band and connected to the posterior lower leg brace and the distal lower leg with a belt. The trunk was supported in a vertical position, and both upper limbs were crossed anteriorly to the trunk. An isometric exercise in which the knee on the non-paralyzed side was maximally extended for approximately 3 s was then performed. Next, an isometric exercise of maximum knee extension on the paralyzed side was performed for approximately 3 s. Both sides were measured twice each. The values obtained were normalized by the distance of the lateral epicondyle of the knee to the sensor pad and the body weight. The mean values of the two recordings for each participant were used [20].

#### 2.3.3. Physical Performance

The SPPB is a composite outcome measure of lower limb function, including strength, endurance, gait, and balance [21]. The SPPB measures three components: walking speed, chair stand, and standing balance. Each task of the SPPB was assigned a score ranging from 0 to 4 (0 = inability to complete the task; 4 = highest level of function), and the sum of these three tasks (0–12) reflects the complete measurement of physical function.

To assess composite walking ability, we performed the TUG test. The TUG test is used in clinical practice to evaluate functional ambulatory mobility or dynamic balance in patients. The participants were given the signal “ready, 1, 2, 3, and go”. On the go cue, the participants stood up, walked 3 m, turned around a mark on the floor, walked back, and sat down [22]. This assessment has also been used to evaluate the risk of falling in patients with stroke [23,24].

The ability to maintain postural stability was measured using a stabilometer function in BASYS. The participant stood on the device, maintaining a standing posture with both hands on the body and trying to keep steady as much as possible. Two measurements of 30 s were performed in this study. The length of the COP trajectory was collected at a sampling frequency of 1000 Hz, and then the analysis software in the device calculated the trajectory length per second (cm/s) by dividing the total length by the total time of the test.

#### 2.3.4. Self-Reported Questionnaires for Fall and Gait Efficacy

Two questionnaires, the modified Gait Efficacy Scale (mGES) and the Fall Efficacy Scale (FES), were used to assess changes in individuals’ perceptions of walking due to the intervention. The mGES is a self-report 10-item scale of cognitive confidence in walking ability; individual items on the mGES were rated from 1 (no confidence) to 10 (full confidence). Items represented a variety of tasks ranging from level walking to walking on uneven surfaces, curbs, or stairs [25]. The mGES total score was the sum of the item scores in the range of 10 to 100.

The FES assesses the fear of performing activities necessary for daily living and can measure the degree of confidence in fall injuries during activities [26]. The FES consists of 10 questions and uses a 10-point measure in which each question can be answered with a minimum of 1 point and a maximum of 10 points. Higher points imply greater confidence that one will not be injured in a fall.

### 2.4. Statistical Analysis

Statistical analysis was conducted using SPSS (IBM SPSS Statistics for Windows version 27.0; IBM, Sandi Co., Ltd., Tokyo, Japan). Independent *t*-tests and chi-square tests were used to compare the baseline characteristics between the two groups. A two-way split-plot analysis of variance (ANOVA) with repeated measures was used to analyze the main effect as well as the interaction of group and time for the outcome measures. The effect size for the interaction effect of the two-way repeated ANOVA was calculated using eta-squared (*η*^2^) statistics. When the two-way repeated ANOVA result showed statistical significance, the least significant difference (LSD) method was used for multiple analyses to investigate the difference in values between pre, post, and follow-up. The effect size for the interaction effect of the two-way repeated ANOVA was calculated using *η*^2^ statistics; for the post hoc, LSD comparison was calculated using Cohen’s d statistics, and the observed power of each was generated using the G*Power software. The significance level was set at *p* < 0.05.

The distribution-based approaches were performed to determine the minimum clinically important difference (MCID). The MCID was calculated using the distribution-based Cohen effect size benchmark. An effect size of 0.5 (0.5 SD of the baseline score) indicates a crucial change. In this study, the post-intervention changes were detected by TUG. To assess the extent of change of TUG in patients, we examined the percentage of subjects whose change scores exceeded the values for distribution-based MCID.

## 3. Results

Of the 30 included participants, there was one dropout in the intervention group (declined post-evaluation) and two dropouts in the control group (difficulty with the twice-weekly intervention (*n* = 1) and fall at home leading to becoming an inpatient (*n* = 1)) during the intervention. In addition, three patients in the intervention group and two patients in the control group dropped out during the follow-up period (all of whom voluntarily avoided visiting the day-care center due to the spread of the coronavirus disease). There were no adverse events reported or observed by the participants related to the intervention or measurements, such as a fall, feeling fearful, or becoming unwell, in this study. The flow of this study process is shown in Figure 2. Finally, 11 participants in each group completed the 6-week intervention protocol and were followed up for 10 weeks. The groups did not differ significantly in any of the demographic and clinical characteristics (Table 1).

Table 2 presents the results of comparisons among the groups and within each group assessed at post-intervention and follow-up for each outcome measure. A significant interaction was observed only for the TUG test (*F* = 6.078, *p* = 0.007) and mGES (*F* = 3.759, *p* = 0.038), and no significant interaction was observed for other measures. The observed powers for TUG and mGES were calculated to be 0.999 and 0.994, respectively, which were statistically sufficient for detection.

The results of the post hoc test comparison of the differences in the outcome measures of each period for TUG and mGES are shown in Table 3. In the intervention group, TUG was significantly improved from pre- to post-intervention (*p* = 0.015). In addition, the changes in pre-intervention to follow-up measurements on TUG (*p* = 0.016) and mGES (*p* = 0.036) showed statistical significance. In the control group, the TUG test showed no significant changes for any of the three periods, and mGES decreased significantly from pre-intervention to follow-up and post-intervention to follow-up (*p* = 0.040 and *p* = 0.039, respectively). Power analysis revealed adequate power (80% or greater) to detect clinically important changes in these parameters.

The MCID estimate based on the distribution of TUG was 3.51 s. After the intervention, three patients (27.3%) in the intervention group and no patient (0%) in the control group showed clinically effective changes.

## 4. Discussion

This is the first single-blind and single-center RCT on balance exercise using a real-time postural feedback system. We hypothesized that (1) the intervention effect would be higher than that of the conventional balance exercise on unstable surfaces, and (2) the effect obtained during the intervention period would be sustained after four weeks. The results of this study suggest that a real-time postural feedback system is effective in decreasing time to perform TUG and increasing the efficacy of the patient’s gait compared to common balance exercises using an unstable surface in patients with chronic stroke. Furthermore, the effects of the intervention were maintained after four weeks of follow-up. However, the intervention protocol in this study did not shorten the TUG enough to satisfy the MCID (3.51 s), suggesting that a more effective intervention protocol needs to be developed. Thus, the results partly support the hypothesis.

According to the guidelines for treating patients with stroke in the acute phase, high frequency and high dose of rehabilitation are strongly recommended [27]. The intervention group showed interesting results, although the intervention frequency was only twice a week for six weeks. Postural control depends on somatosensory feedback provided by the foot pressing against the support surface. The difference between the intervention and control groups in this study was only regarding balance exercises with non-sensory movements of the platform or under unstable conditions. Therefore, the results obtained in this study indicate that the balancing exercise of sensory feedback to an extent at a range of non-perceptual levels could help improve walking ability by the participants’ enhanced feedback to the somatosensory and vestibular systems. In a recent study, balance exercises under restricted visual conditions increased somatosensory and vestibular feedback and improved walking ability more than exercises on unstable surfaces [28]. When standing on an unstable surface, the central nervous system is more sensitive to sensory feedback from the visual and vestibular systems and less sensitive to feedback received from the somatosensory system. Contrarily, when visual feedback is restricted, balance control depends almost entirely on feedback from the somatosensory and vestibular systems [11]. Hence, considering the ability to select and use the various sensory contributions of each system in different environments (sensory reweighting) [11,29], the intervention in this study was similar to visual restriction methods. It increased somatosensory and vestibular feedback, which may have resulted in improved TUG performance. Balance exercises on unstable surfaces in this study showed no improvement in physical function. A previous study on unstable surface exercise for six weeks showed improved static and/or dynamic postural control among patients with chronic stroke [8,30]. However, the study involved multiple exercise programs with an intervention frequency of six times per week [8,30]; thus, the difference in intervention frequency and exercise programs might have influenced the control group’s results.

In general, a high-frequency exercise regimen is strongly recommended for rehabilitation in stroke, but the appropriate exercise load for balance exercise is still unclear [27]. In the current study, a low-frequency intervention was effective. Yet another study investigated the effects of balance exercises using a moving floor, and it was found that stance stability improved following intervention [31]. In the current study, the protocol was similar to the one used in that study (Dickstein et al.: three times a week for five weeks, current study: twice a week for six weeks), and it was found that the intervention was effective in improving balance even if the sensory feedback was provided to an extent at a range of non-perceptual levels. The results of this study showed that TUG time was improved by balance exercises with non-sensory gravitational sway. However, the MCID was 3.51 s in the current study, which was achieved by only three patients in the intervention group. Compared to other studies, the improvement in this study was greater than the standard error of measurement (1.14 s) but less than the smallest real difference (23% change) [32]. Therefore, we would like to conduct further studies to validate a more effective intervention program, such as increasing the frequency of interventions, and to provide a safer way to introduce balance exercises to patients with stroke, elderly patients with chronic stroke, and patients with a history of falls.

The key results of this study were as follows: (1) the TUG improved after the intervention; (2) this improvement was maintained after four weeks of intervention; and (3) the participants gained more self-confidence in their gait performance. A previous report showed that mGES is associated with TUG and walking speed [25]. This suggests that the improvement in gait ability during the intervention period increased the participant’s self-confidence in gait performance, and the continuance of the effect might have resulted in a significant increase in self-confidence after four weeks of intervention. In fact, the fine-tuning network of locomotion is relevant when considering the improvement of gait for immediate and long-term gait adaptation. This concerns not only rehabilitation but also the frequency of walking after returning home. In order to facilitate the achievement of optimal gait function, it is important that subjects are comfortable or willing to walk; the importance of this willingness has been shown in previous studies [33]. The detailed mechanism of the improvement in the TUG in this study is unknown, but a previous study showed that the effect of the anti-phase mode of the same device as in the present study for healthy adults reduced the onset time of muscle activity in response to changes in COP oscillations [17]. Elderly people with stroke have delayed onset of posture reflex muscle onset time in the paralyzed limb, and it is one of the factors that reduces postural control [34]. Therefore, a similar mechanism for improvement in postural control might have been caused in the patient after the stroke, resulting in a shorter TUG that required switching movement between rising from a chair, walking, turning, and sitting down. In addition, we hypothesize that the patient’s postural control (muscle coordination ability in this study) improved, which in turn increased the patient’s confidence and motivation to walk. Moreover, the continuous act of walking at home, besides rehabilitation, might have led to the patient’s improved self-confidence.

This study has some limitations. First, there are risks for bias. This study was conducted at a single center, and the results of the study were based on patients in a day-care center, which might have caused selection bias. In addition, because this was a single-blind study, the physical therapists who conducted the intervention and assessment knew the balance exercise conditions and interventions. To minimize the bias in this study, a narrative document was created to provide uniform information to patients during intervention and assessment. However, it is unclear whether this minimized all risks of bias in the outcome of the assessment. Second, although the difference of the baseline score for the TUG was not statistically significant, it still needs to be considered. This means that the specifics of whether the participants in the intervention group of this study would benefit from the use of a polyurethane mat, or if the control group would benefit from the use of the stabilometer with in-built disturbance generation, needs more discussion. Future studies with larger sample sizes and those that include TUG time as an adjustment factor in the allocation need to be considered. Additionally, it is necessary to conduct a double-blind study at other institutions. Third, this study focused on patients with chronic stroke. In physical therapy after stroke, rehabilitation is also performed during the acute and recovery phases. This exercise is safe to carry out even in the acute and recovery phases; thus, we believe that it could be successfully applied to rehabilitation in other phases. Fourth, the results of this study do not show the effect of balance exercises alone since they were performed in addition to regular physical therapy. Nevertheless, this study provides clinically valuable information because the results showed a higher improvement in the test group than in the control group, and the balance exercises were used in addition to the usual physical therapy in actual clinical practice. In the future, we need to continue to study intervention protocols that satisfy the MCID and provide better information for clinical practice. The detailed mechanism of improvement requires further elaboration, but the fact that the exercises could be performed safely and effectively is important for both patients with chronic stroke and practitioners as this can help prevent secondary injuries such as falls.

## 5. Conclusions

In the current study, we found that balance exercises with unnoticed platform movements could improve the walking ability of patients with chronic stroke and increase their self-confidence in gait performance. In addition, the effects of the exercises were maintained even after four weeks, which is clinically valuable as a method to safely improve the balancing ability of patients with chronic stroke. In the future, we will carry out further studies using intervention programs that bring about clinically valuable changes.

## Figures and Tables

**Figure 1 brainsci-11-01493-f001:**
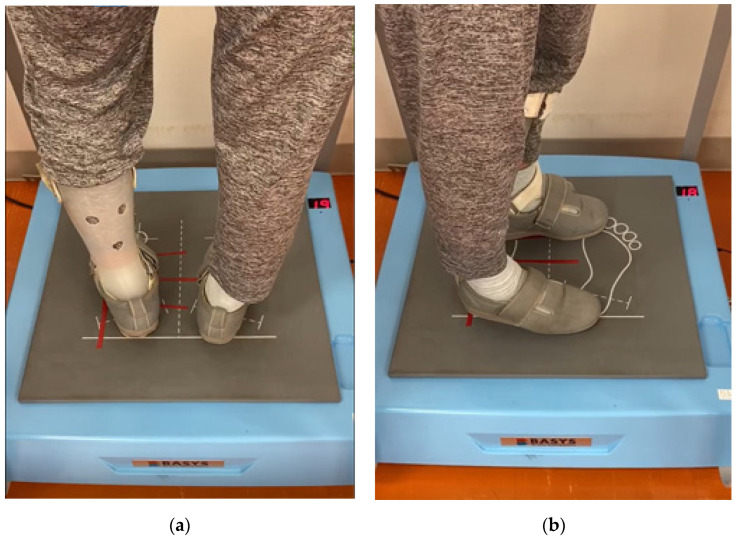
The foot position during balance exercise using real-time feedback system (BASYS). (**a**) Anterior-posterior direction exercise. (**b**) Medial-lateral direction exercise.

**Figure 2 brainsci-11-01493-f002:**
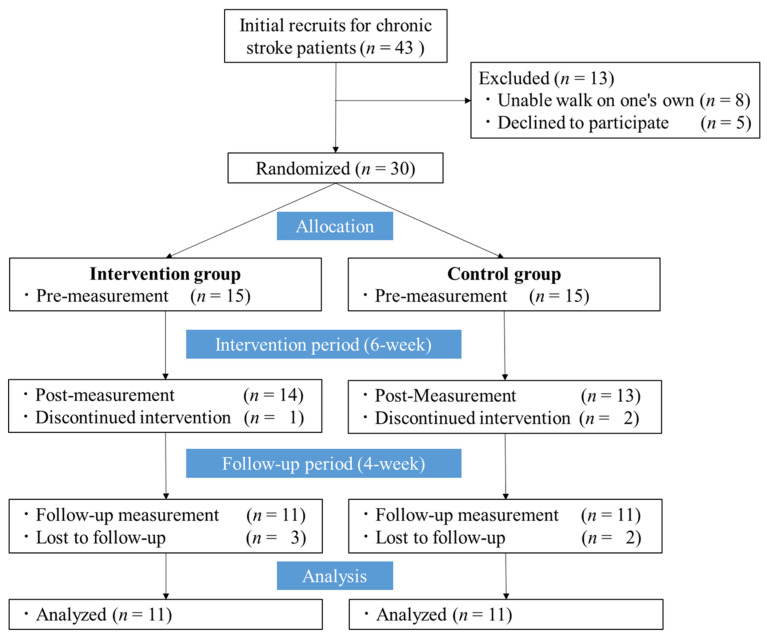
Time course of the study.

**Table 1 brainsci-11-01493-t001:** Participants’ characteristics at analysis.

Variables	Total (*n* = 22)	Intervention Group (*n* = 11)	Control Group (*n* = 11)	*p*-Value
Age (years)	75.0 ± 11.5	73.6 ± 12.5	76.4 ± 10.9	0.591
Height (cm)	158.6 ± 9.9	158.6 ± 10.7	158.7 ± 9.4	0.983
Weight (kg)	61.4 ± 8.9	60.6 ± 8.8	62.2 ± 9.4	0.687
BMI (kg/m^2^)	24.5 ± 3.5	24.2 ± 3.2	24.8 ± 3.9	0.677
Sex: Female (*n*, %)	7 (31.8)	3 (27.2)	4 (36.4)	0.901
Use of walking equipment (*n*, %)	16 (72.7)	9 (81.8)	7 (63.6)	0.580
Use of ankle foot orthosis (*n*, %)	4 (18.2)	3 (27.3)	1 (9.1)	0.228
Duration after stroke (months)	43.0 (24.0–85.5)	42.0 (26.0–138.0)	66.0 (20.0–72.0)	0.656
HDS-R (points)	24.1 ± 5.4	24.9 ± 6.3	23.4 ± 4.7	0.520
Brunnstrom Recovery Stage (Lower Limb)	III:3, IV:3, V:6, VI:10	III:1, IV:2, V:4, VI:4	III:2, IV:1, V:2, VI:6	0.630

Values are reported as mean ± standard deviation (SD). Only duration after stroke (months) reported as median and interquartile range (IQR). BMI: body mass index, HDS-R: revised Hasegawa’s dementia scale. Brunnstrom recovery stages include I to VI, and the larger number indicates better development of motor functions and reorganization of the brain after a stroke.

**Table 2 brainsci-11-01493-t002:** Pre (0-week), post (6-week), and follow-up (10-week) outcome measures of each group.

Variables	Intervention Group (*n* = 11)	Control Group (*n* = 11)	Main Effect				Interaction Effect				
			Time		Group		Time * Group				
	Mean ± SD	Mean ± SD	*F*	*p* Value	*F*	*p* Value	*F*	*p* Value	*η* ^2^	Effect Size	Observed Power
Muscle strength of knee extension (Nm/kg) NPA			0.525	0.547	0.015	0.905	0.153	0.800	0.008	0.090	0.132
Pre	1.52 ± 0.41	1.47 ± 0.35									
Post	1.54 ± 0.26	1.56 ± 0.46									
Follow-up	1.55 ± 0.18	1.52 ± 0.44									
Muscle strength of knee extension (Nm/kg) PA			0.147	0.862	0.011	0.919	0.351	0.705	0.017	0.132	0.237
Pre	1.24 ± 0.52	1.25 ± 0.50									
Post	1.25 ± 0.53	1.20 ± 0.51									
Follow-up	1.24 ± 0.48	1.22 ± 0.45									
SPPB score (points)			1.625	0.215	0.876	0.360	0.768	0.443	0.037	0.196	0.478
Pre	8.00 ± 3.46	9.45 ± 2.46									
Post	8.36 ± 3.47	9.18 ± 1.94									
Follow-up	8.73 ± 2.94	9.64 ± 1.80									
TUG (s)			5.004	0.015 ^♀^	1.944	0.178	6.078	0.007 ^♀^	0.233	0.551	0.999
Pre	16.95 ± 8.28	11.97 ± 4.66									
Post	15.01 ± 6.54	12.31 ± 4.80									
Follow-up	14.71 ± 6.29	11.93 ± 4.10									
COP length (cm/s)			0.263	0.770	0.682	0.419	0.068	0.934	0.003	0.054	0.079
Pre	2.33 ± 1.99	1.78 ± 0.96									
Post	2.24 ± 1.95	1.77 ± 0.81									
Follow-up	2.23 ± 1.75	1.71 ± 0.83									
mGES (points)			0.457	0.615	2.038	0.169	3.759	0.038 ^♀^	0.158	0.433	0.994
Pre	46.27 ± 19.74	63.36 ± 19.56									
Post	52.64 ± 14.19	63.36 ± 17.02									
Follow-up	56.82 ± 17.68	55.91 ± 14.65									
FES (points)			2.387	0.106	0.707	0.410	0.248	0.777	0.012	0.110	0.177
Pre	296.36 ± 50.45	317.27 ± 60.18									
Post	306.36 ± 62.17	327.27 ± 46.71									
Follow-up	290.00 ± 53.67	298.18 ± 51.54									

PA: paralyzed; NPA: non-paralyzed; SPPB: short physical performance battery; TUG: timed up and go test; COP: center of pressure; mGES: modified gait efficacy scale; FES: fall efficacy. ^♀^ A significant difference between intervention group and control group.

**Table 3 brainsci-11-01493-t003:** Difference for the outcome measures between the three periods.

Variables	Intervention Group (*n* = 11)				Control Group (*n* = 11)			
	Difference [95% CI]	*p* Value	Effect Size	Observed Power	Difference [95% CI]	*p* Value	Effect Size	Observed Power
TUG test (s)								
Post–Pre	−1.95 ± 0.660 (−3.42, −0.473)	0.015 ^a^	−0.888	0.977	0.34 ± 0.26 (−0.25, −0.92)	0.227	0.388	0.411
Follow-up–Pre	−2.24 ± 0.77 (−3.97, −0.52)	0.016 ^b^	−0.875	0.974	−0.04 ± 0.39 (−0.92, 0.83)	0.912	−0.034	0.052
Follow-up–Post	−0.30 ± 0.52 (−1.45, 0.86)	0.855	−0.173	0.121	−0.38 ± 0.36 (−1.19, 0.42)	0.313	−0.320	0.299
mGES (points)								
Post–Pre	6.36 ± 6.25 (−7.57, 20.30)	0.330	0.307	0.279	0.00 ± 3.85 (−8.57, 8.57)	1.000	−0.591	0.752
Follow-up–Pre	10.55 ± 4.36 (0.84, 20.26)	0.036 ^b^	0.730	0.903	−7.46 ± 3.16 (−14.48, −0.43)	0.040 ^b^	−1.365	0.999
Follow-up–Post	4.18 ± 6.35 (−9.96, 18.32)	0.525	0.199	0.144	−7.46 ± 3.14 (−14.45, −0.46)	0.039 ^c^	−1.369	0.999

TUG, timed up and go test; mGES, modified gait efficacy scale. ^a^ Significant difference between pre- and post-intervention (*p* < 0.05). ^b^ Significant difference between follow-up and pre-intervention (*p* < 0.05). ^c^ Significant difference between follow-up and post-intervention (*p* < 0.05).

## Data Availability

The datasets used and/or analyzed during the current study are available from the corresponding author on reasonable request.

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
