# Peer review of "Effect of 6-Week Balance Exercise by Real-Time Postural Feedback System on Walking Ability for Patients with Chronic Stroke: A Pilot Single-Blind Randomized Controlled Trial"

_brainsci, 2021, doi:10.3390/brainsci11111493_

Round 1
Reviewer 1 Report
Comments regarding manuscript
“Effect of 6-week balance exercise by real-time postural feedback system on walking ability for patients with chronic stroke: a pilot single-blind randomized control trial”
The authors performed a single-blind RCT in 22 participants (8 participants didn’t complete the study protocol). The aim of the study was to evaluate the effect of a balance training based on a “real-time postural feedback system (Balance Adjustment System)” on muscle strength, physical performance, gait, and balance after 6 week of training and after 4 weeks from the end of the training protocol (follow up assessment). The main results presented by the authors were: a) The 6-week balance training protocol based on a “real-time postural feedback system” improved performance in TUG and in self-confidence of walking ability (m GES).
Mayor comments
Introduction:
- It is not clear the potential connection between this “real-time postural feedback” stimulation and walking speed or walking performance. In the introduction the authors should state the potential mechanisms that could explain why an intervention like the one that the authors propose could have benefit on gait (results in a previous work on patients with Parkinson disease is not enough).
- I suggest keeping the aims and hypothesis simple and include the second aim in the first one. I understand that to see if the potential changes induced by the intervention protocol last until the follow up assessment could be a plausible second question itself, however, in my opinion, include this in a global aim regarding the effect of the intervention at the end of the training protocol and at the follow up assessment is simpler.
- If the participants don’t feel the oscillation of the platform, is it correct to name the intervention device as a “postural feedback system”?.
Methods:
- The principal outcome measure is the Timed up and go, a clinical outcome measures very common in the rehabilitation and clinical field. I would like to ask the authors to report not only the statistical significance, but also the minimal clinical important differences (MCID), which determine the minimal amount of change that is important to the patient. And use these results to discuss the potential benefits of the proposed intervention.
- A better justification should be given regarding how a training protocol that include 4 minutes of practice per week can induce long term changes on walking performance in persons with stroke.
Discussion:
- Sensory reweighting has been described as the process of adjusting the sensory contributions to improve balance control. It is not clear how individuals could experience a process of sensory reweighting if the stimulation provided by the platform is not filed by the participant. I would like to ask the authors to elaborate more this part of the discussion and provide more evidence (published evidence) regarding how an unperceived sensory stimulation could impact on the sensory reweighting process and generate functional changes.
Minor comments:
- Please correct all the grammar errors and typos mistakes.
- Page 3, line 122. Please do not cite unpublished data. You could present those date in a table as a preliminary data.
- Please provide more technical details about how the “Balance Adjustment System” works, such as frequency of oscillation, the justification of the amount of oscillation and the justification of the amount of training time that is suggested.
Author Response
Dear Reviewer 1,
Thank you for inviting us to submit a revision of our manuscript.
We appreciate the time and effort that you have dedicated in providing insightful feedback on our manuscript. In addition, our manuscript had many mistakes. We apologize for the mistake. After incorporating the changes suggested, we have sent paper for English editing once more.
Based on your suggestions, we have incorporated changes into the manuscript and now hope that this manuscript addresses all previous concerns that were noted.
The attached PDF file is the manuscript with the corrections you have pointed out and highlighted in yellow lines.
Again, thank you for your kind review. We hope that these revisions persuade you to accept our submission.
Sincerely,
Authors

Reviewer 2 Report
1. In the introduction: "The most common physical dysfunction related to chronic stroke is impaired balance (approximately 87.5%)"
Authors can try to mention the range of balance impairment with more references, however its not just approximately 87.5%
2. Balance exercises for patients with chronic stroke include the use of unstable surfaces [6] and virtual reality [7].
There are many other balance rehabilitation strategies which could be mentioned here with appropriate references
3. Change the use of stroke patients to patients with stroke through out the manuscript
4. In methodology, assessment of knee extension strength, the authors have tested the knee extension strength of the non paralysed limb..? needs clarification
5. How the authors blinded the study participants is not explained, needs explanation for the blinding..
6. The authors have used the power analysis after the intervention too., however to make sure whether the authors have used the intention to treat analysis for the drop outs in the follow up.
7. Though there was no notable effect size in the TUG in pre – follow up and post – follow the mGES scores had a large effect size. Could authors give justification for that as both the outcome measures are measuring similar domains.
8. In discussion: "the intervention in this study was similar to visual restriction methods in that it increased somatosensory and vestibular feedback, which may have resulted in improved TUG performance".
How did the authors manage to give visual restriction to the subjects.
9. The results of the study are also due to the reactive postural control mechanism, which could have been highlighted in the discussion.
Author Response
Dear Reviewer 2,
Thank you for inviting us to submit a revision of our manuscript.
We appreciate the time and effort that you have dedicated in providing insightful feedback on our manuscript. In addition, our manuscript had many mistakes. We apologize for the mistake. After incorporating the changes suggested, we have sent paper for English editing once more.
Based on your suggestions, we have incorporated changes into the manuscript and now hope that this manuscript addresses all previous concerns that were noted.
The attached PDF file is the manuscript with the corrections you have pointed out and highlighted in yellow lines.
Again, thank you for your kind review. We hope that these revisions persuade you to accept our submission.
Sincerely,
Authors

Round 2
Reviewer 1 Report
Comments regarding manuscript
“Effect of 6-week balance exercise by real-time postural feedback system on walking ability for patients with chronic stroke: a pilot single-blind randomized control trial”
Most of the comments were well addressed by the authors. However, I still have some concerns regarding the concept of sensory feedback and sensory reweighting in a context in which the proposed device generates a stimulus imperceptible for the patients. I understand that you can sense or register some reflex response after the imperceptible stimulus triggered by the interventional device, however, when you talk about sensory feedback, or sensory reweighting, there are cognitive component involve in this process. In this context, according to my understanding, the authors should explain better how the participants experience a process of sensory feedback and specially a process of sensory reweighting without perception.
Additionally, after to see the MCID results for the principal outcome measures, the authors should clarify that the results of the protocol are not very strong, and the hypothesis is partially responded.
Author Response
Dear Reviewer 1,
Thank you for inviting us to submit a revision (Round 2) of our manuscript.
We appreciate the time and effort that you have dedicated in providing insightful feedback on our manuscript.
Based on your suggestions, we have incorporated changes into the manuscript and now hope that this manuscript addresses all previous concerns that were noted.
The attached PDF file is the manuscript with the corrections you have pointed out and highlighted in yellow lines.
Again, thank you for your kind review and opportunity to improve the manuscripts. We hope that these revisions persuade you to accept our submission.
Sincerely,
Authors

Reviewer 2 Report
The authors have modified and clarified all the comments in a satisfactory manner.
Author Response
Dear Reviewer 2,
Thank you for reviewing the revisions of our manuscript.
Your suggestions have helped us to revise the manuscript to improve it.
Again, thank you for your kind review and opportunity to improve the manuscripts.
Sincerely,
Authors
This manuscript is a resubmission of an earlier submission. The following is a list of the peer review reports and author responses from that submission.
Round 1
Reviewer 1 Report
Thank you for the opportunity to review this randomised controlled trial. In this study a the effect of using a real time postural feedback system on balance in chronic stroke survivors is investigated.
Abstract
Page 1 line 26: I suggest replacing “Shortening in TUG” with “decreased time to perform TUG” or “improved TUG’
Page 1 line 20: “Thirty participants were divided…” replace with Thirty participants were randomized…”
The results show improvements in TUG and mGES (self-reported gait efficacy) based on this can you conclude that walking ability improved?
Introduction
Page 2, line 46-50: The connection between the effects found of “underwater” (water-based?) therapy on balance and the use of a feedback system is not clear to me, could you elaborate on this a bit more. I suggest discussing the effects of feedback in other types of therapy for stroke to develop the rationale for using a feedback system to improve balance after stroke.
Page 2 line 64-66: Please specify what the outcome of interest (i.e. the main outcome) is in the aims.
Materials and Methods
Page 2 line 73: single blind please specify who was blinded (i.e. patients, assessors, data-analyses)
Page 4 please specify the primary outcome
Page 2 line 81-82: Please specify which effect sizes for which outcome (primary outcome?) were used to calculate the sample in G power?
Inclusion criteria: it seems that the walking ability of include patients was quite high based on the inclusion criteria (i.e. able to walk 10 metres independently without a device), what was the reason for this criteria. Additionally, were there any criteria regarding balance impairments as this was the main focus of the intervention?
Results
The SD of the time since stroke indicate the data for this variable is skewed, reporting time since stroke in medians and IQR or range would be more informative.
Were adverse effects of the intervention monitored and were any reported?
Were participants asked if they could feel the platform move?
Was usual care monitored, i.e type, frequency, intensity and if so was it different between groups? I think it is important to discuss how activities during usual therapy and outside therapy time between groups might have an impact on the results.
What do the scores on the different outcomes mean, consider reporting additional detail about clinical relevance of the results.
Page 7 line 218-19: “…this hypothesis” explicitly state the hypothesis.
Page 7 line 220: Unclear, I assume that this sentence speaks to the surprising results given that the “dose of the intervention was quite low”, consider rewriting this sentence.
The previous point about the frequency and duration of the intervention (1-2 min, two times per week for six weeks) needs a bit more consideration in the discussion. Currently the evidence for effective stroke rehabilitation seems to point to towards high intensity, large dose intervention (i.e. large number of repetitions), which is contrary to the findings of this study.
Please discuss any biases that might have impacted on the results of this study.
Conclusions
Page 8 line 263-54: Unclear, please explain why and how interventions should be chosen carefully?
Author Response
Dear Reviewer 1,
Thank you for inviting us to submit a revision of our manuscript.
We appreciate the time and effort that you have dedicated in providing insightful feedback on our manuscript.
Based on your suggestions, we have incorporated changes into the manuscript and now hope that this manuscript addresses all previous concerns that were noted.
The attached PDF file is the manuscript with the corrections you have pointed out and highlighted in yellow lines.
Again, thank you for your kind review. We hope that these revisions persuade you to accept our submission.
Sincerely,
Authors

Reviewer 2 Report
This study is considered as a study to examine the effect of balance training using a real time postural feedback system for 6 weeks on the recovery of balance ability in stroke patients. In particular, it is considered as a study to present a quantitative effect by comparing the training effect with the control group. This intervention is considered to be one of the clinically applicable intervention methods to improve the balance ability of stroke patients. However, it is difficult to find the difference between the training in this study and the balanced training method in previous studies. Balance training using equipment, especially equipment that can induce consistent movement on the ground, has already been actively conducted in a number of previous studies. Its effectiveness has also been proven and is a generalized intervention method. In addition, this RCT study was divided into an intervention group and a control group to compare the intervention effects. Nevertheless, it is judged that the measurement of postural control, a major variable in this study, provided a more favorable environment for the intervention group. Since the intervention group performed training/evaluation in the same environment using BASYS equipment, the learning effect is expected to be greater than that of the control group. Finally, the sample size was calculated to perform the RCT study. How much effect size did you enter? Has the sample size been calculated including follow-up tests? Compared to the calculated sample size, the final sample size for the study is small. Therefore, it is judged that the power of the results of this study is somewhat lower.
Title
- The dependent variable is missing from the study title. You need to add a dependent variable.
Abstract
- A description of the main results based on the data analysis method of this study is required. Please present the significance of the interaction for group*time and the results of the post-hoc analysis.
Introduction
- Lines 41-50: This study is considered to have been conducted to prove the training effect using somatosensory feedback by ground motion. Therefore, it is necessary to review prior research related to feedback. For example, balance training interventions using visual and auditory feedback,,,
- Lines 61~63: There are many intervention studies using a real time postural feedback system. In particular, the Biodex-balance system is an optimized system and there are many related studies. Therefore, the originality of this study is questionable.
Material and Methods
- It is necessary to present the criteria for selection and exclusion of subjects in one paragraph. In the study results, demographic data for users of foot orthosis and walking aids were presented. However, it is questionable whether the influence of the use or absence of orthosis on the intervention effect was not considered even within the same group.
Intervention
- The training time is somewhat disappointing compared to the training application period. 8 minutes of relevant training application per week?
- Training difficulty adjustment has been increased by 5%. Were there any cases of patients who could not perform the exercise at increased intensity? Did all patients complete the procedure? How did you prevent a fall down that could occur during training?
- The training applied to the intervention group was judged as ground movement on a horizontal plane. The training applied to the control group is judged as ground motion induced in the 3D plane. Are these motion vectors judged to be the same? Also, is it reasonable to use the term real time only in the intervention group between the two training methods?
Outcome measure
- Is there a reason why only the knee extensor was used to measure muscle strength? Reference required. Also, it is difficult to understand why only the muscle strength of the non-paralyzed side was measured. Shouldn't the paralytic side muscle strength be measured in patients with reduced balance ability?
- BAYSYS was used to measure postural stability. Don't you think it is a favorable measurement condition for the experimental group?
- I would like to know why the LSD was performed for the post-hoc analysis. Shouldn't a comparison between groups be performed on the interaction, that is, the amount of change before and after the intervention?
Results
- It is judged that the influence of the results has decreased somewhat due to the decrease of the sample size.
- It is necessary to present the results of comparison between groups for the Brunnstrom Recovery Stage.
- It is necessary to describe the results of post-hoc analysis of interactions.
Discussion
- It is necessary to focus on the results of the post-hoc analysis, that is, the results of comparison between groups on the amount of change.
Author Response
Dear Reviewer 2,
Thank you for inviting us to submit a revision of our manuscript.
We appreciate the time and effort that you have dedicated in providing insightful feedback on our manuscript.
Based on your suggestions, we have incorporated changes into the manuscript and now hope that this manuscript addresses all previous concerns that were noted.
The attached PDF file is the manuscript with the corrections you have pointed out and highlighted in yellow lines.
Again, thank you for your kind review. We hope that these revisions persuade you to accept our submission.
Sincerely,
Authors

Round 2
Reviewer 1 Report
Please provide additional details for the HDS-R scale was a cut-off point used for inclusion what do the scores mean and the same for Brunnstrom recovery stages.
Effect size was 0.25 for the power calculation. Please provide a rationale for that effect size and incl reference to any relevant studies.
Exclusion criteria: please remove duplicate exclusion criteria mentioned on page 2 line 93 and 94
To me it is unclear what type of patients were eligible for the study. From the criteria described it is not clear why you have as many patients included with walking/balance impairments. Maybe describe what “type” of patients come to the day care centre (i.e. why do they come to the centre is it for ongoing therapy, do they have ongoing impairments).
Results
Please comment on adverse events were they monitored and report if there were none or any reported by participants.
How did the therapists check if patients could feel the platform move or not?
The baseline scores on the TUG seem different between the two groups have you considered adjusting from this difference. Also, I think it would be appropriate to discuss the implications of the difference on the results of this study.
Additionally, please discuss if the difference between the mean TUG scores of the intervention and control group is clinically meaningful.
Author Response

(The authors gave the same response as above.)

Reviewer 2 Report
Revision requirements were well-received overall.
Author Response
Dear Reviewer 2,
Thank you for reviewing the revisions of our manuscript.
Your suggestions have helped us to revise the manuscript to improve it.
Again, thank you for your kind review and opportunity to improve the manuscripts.
Sincerely,
Authors